# Data-Driven Approach to Encoding and Decoding 3-D Crystal Structures

## Abstract

Generative models have achieved impressive results in many domains including image and text generation. In the natural sciences, generative models have lead to rapid progress in automated drug discovery. Many of the current methods focus on either 1-D or 2-D representations of typically small, drug-like molecules. However, many molecules require 3-D descriptors and exceed the chemical complexity of commonly used datasets. We present a method to encode and decode the position of atoms in 3-D molecules along with a dataset of nearly 50,000 stable crystal unit cells that vary from containing 1 to over 100 atoms. We construct a smooth and continuous 3-D density representation of each crystal based on the positions of different atoms. Two different neural networks were trained on a dataset of over 120,000 three-dimensional samples of single and repeating crystal structures. The first, an Encoder-Decoder pair, constructs a compressed latent space representation of each molecule and then decodes this description into an accurate reconstruction of the input. The second network segments the resulting output into atoms and assigns each atom an atomic number. By generating compressed, continuous latent spaces representations of molecules we are able to decode random samples, interpolate between two molecules, and alter known molecules.

## 1 Introduction

Generative models have recently seen tremendous success in generating 2-D images of every day objects (Kingma & Welling, 2013; Goodfellow et al., 2014; Brock et al., 2018; Razavi et al., 2019). The size and accuracy of generated results has greatly improved to the point where samples from the latent space decode to photo-realistic samples (Brock et al., 2018; Razavi et al., 2019). A very exciting and important future avenue for generative models is the generation of 3-D structures, like in the world around us. Adversarial networks and autoencoders have been extended into 3-D and have shown they are able to encode useful representations of everyday objects (Wu et al., 2016; Zhu et al., 2018; Brock et al., 2016; Achlioptas et al., 2017; Valsesia et al., 2018; Meng et al., 2019; Maturana & Scherer, 2015). Representations using point clouds have gained popularity as a way to capture the distribution of different types of objects (Achlioptas et al., 2017; Yang et al., 2019; Wu et al., 2018).

As machine learning approaches are able to understand and recreate the underlying distributions for many different types of objects in our world, the application of these tools in the physical sciences is a very exciting direction. A clear field where generative models can have a tremendous impact is in material discovery, which is very important for designing new batteries or carbon capture devices for fighting climate change.

One very successful area of applying generative models to the sciences has been the field of drug discovery (Jin et al., 2018; Gómez-Bombarelli et al., 2018; Assouel et al., 2018) (see Schwalbe-Koda & Gómez-Bombarelli (2019) for an excellent summary of many methods). The success of machine learning in this domain has been enormously helpful as a process that was often painstakingly slow has been rapidly accelerated in searching an unimaginably large space of possible drug compounds, estimate to be up to $10^{60}$ (Polishchuk et al., 2013). Additionally, excellent datasets such as QM9 (Ramakrishnan et al., 2014) and ZINC (Irwin et al., 2012) which contain molecules of interest along with their precomputed properties have allowed for comparing different methods and developing state-of-the-art tools.

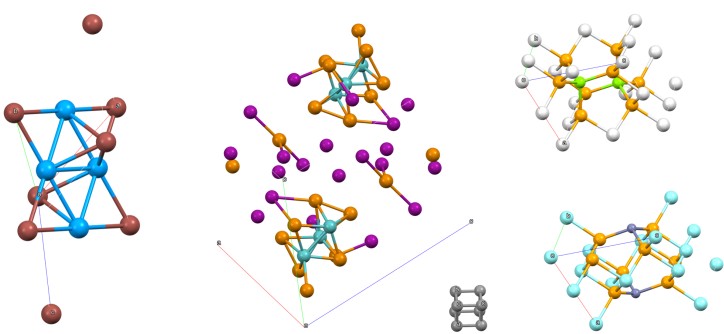

Figure 1: **Examples of crystal unit cells**. Each example shows the unit cell of a crystal. Different colors represent different atomic species. A red, green, and blue line represent the 3 axes of the crystal. Note that they vary in length and angle. Additionally, some unit cells have just one or two atoms while others have nearly 100. Visualizations were made with Mercury (Macrae et al., 2008).

By learning a compressed, useful, latent space representation, this enormous space of molecules can be embedded in a simpler latent space that is easier to search Kusner et al. (2017). Using generative models, compounds hypothesized to have specific properties can be rapidly generated and then a targeted subset of these can be experimentally tested. For example, in Gómez-Bombarelli et al. (2018), an auxiliary property prediction task is introduced for a network separate from the encoder/decoder. This allows for optimization of properties in the latent space and then the resulting latent space vector is decoded into a candidate molecule that can undergo more rigorous computational testing before it is experimentally synthesized. Using a graph representation and reinforcement learning, You *et al* propose a novel method for molecule generation with specific properties You et al. (2018).

Most of the work combining generative models with chemical discovery has focused on molecules that can be represented in either 1 dimension (such as a `SMILES` string (Weininger, 1988)) or by leveraging a 2 dimensional representation of a molecule, such as a graph. **While 1-D and 2-D representations have been very successful for many drug compounds, these representations are not sufficient to describe all molecules.** For example, it is possible for different molecules, with very different properties, to have identical graphs (Gebauer et al., 2019). Additionally, there are more complex classes of compounds where the 3-D structure is integral to the molecule's properties. **Therefore, in this work we focus on generating representations of molecules where the 3-D information is essential for reconstruction.** Generating 3-D structures is still a relatively nascent field, when compared to image and text generation. The data requirements for modeling 3-D structures are larger than their 2-D counterparts and there are fewer standard datasets (Nguyen-Phuoc et al., 2019). Additionally, to measure the properties of inorganic materials (like those considered here), costly DFT calculations are needed which require precise 3-D locations. There is an extra dimension in 3-D problems, providing additional symmetries that often need to be learned (Weiler et al., 2018). New work from generates adjacency matrices which are, by construction, invariant to rotation Hoffmann & Noé (2019). In either case, for the automatic generation of viable complex 3-D structures of inorganic compounds one needs to accurately generate the locations of many atoms in 3-D space.

In this work, we focus specifically on proposing an effective 3-D representation for this class of molecules that can not be represented in the same ways as many smaller molecules. We hope to promote work in this still nascent field which has yet to see the same level of success as the generation of smaller molecules. In the Appendix, we have provided a clear comparison with 1-D and 2-D representations. Currently, no direct comparisons are possible.

In addition to drug discovery, a promising avenue for molecular design is for minimizing environmental impact through the design of more efficient or more environmentally friendly compounds for a variety of applications. Designing more efficient materials for photo-voltaic panels and more environmentally friendly materials for batteries are both very exciting research directions for using machine learning as a tool towards mitigating climate change (Niu et al., 2015; Tabor et al., 2018; Gebauer et al., 2019; Rolnick et al., 2019). Many of the compounds of interest are crystal structures

made of single small sub-units that repeat in all directions. This sub-unit is referred to as a "unit cell" and can vary in size and shape as well as internal arrangement and chemical composition.

In this work, we present a method for encoding and decoding a very important set of 3-D molecules that can not be represented using standard methods used for drug-like molecules. We directly encode and decode 3-D volumes of density from two different data representations. In the first, we use single unit cells, as shown in Fig. 1. Each unit cell is centered in the cube, but randomly rotated. In the second representation, we repeat the unit cell along all three axes such that the resulting sampled cube contains repeated unit cells of crystal structures, like those shown in Fig. 2 (and Fig. S6C). Many very interesting and important molecules, such as materials for solar panels or batteries, are composed of crystalline units. We train a variational autoencoder (VAE) and a network to segment the decoded output based on the true locations of atoms in tandem. By coupling these two tasks, we are able to accurately encode and decode 3-D atomic positions and species. As far as we are aware, this is not an explored direction as a means to represent molecules for encoding and decoding their 3-D arrangement. We summarize our contribution as follows:

- We propose a method for encoding and decoding 3-D molecules that can not be represented by traditional string or graph based approaches. By jointly training a VAE and a segmentation network on the output of the decoder, we train the entire network in an end-to-end fashion.

- We consider two different problems: (1) encoding and decoding single unit cells and (2) encoding and decoding repeated unit cells. In both cases we accurately reconstruct the locations of atoms. We also achieve good results in atomic species identification and future work will improve this front. By sampling from $\mathbf{z} \sim \mathcal{N}(0, 1)$, we generate complex structures that have physically realistic spacing between atoms.

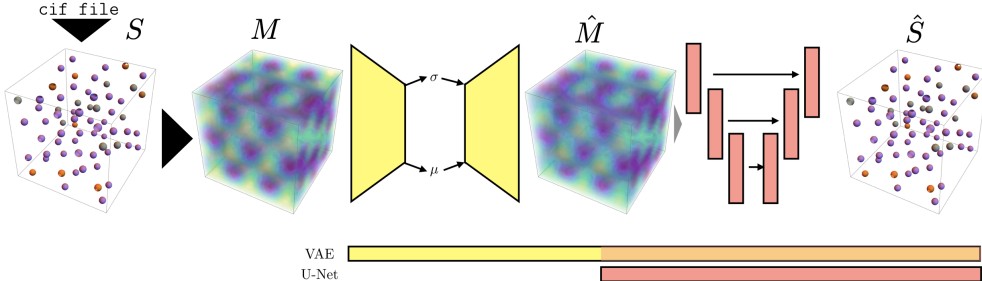

Figure 2: **Network Architecture**. We encode and decode a $30 \times 30 \times 30$ voxel grid representing 10 Å on each side. Each voxel contains the value of the density. The output of the decoder is passed into a 3-D U-Net Çiçek et al. (2016). We train the two models in parallel. In the schematic, we show the crystal represented as a repeated unit cell rather than a single unit cell. The black arrows indicate deterministic transformations. From the `cif` file, the species matrix is constructed. From this, the density matrix is computed.

## 2   PRELIMINARIES AND RELATED WORK

There are many classes of compounds that are of interest for data-driven discovery. There has been significant work exploring the generation of molecules with potential medicinal properties. These molecules tend to be organic molecules and can be represented efficiently using a string or a graph. A more complex class of molecules are inorganic crystal structures which vary in complexity across many different axes. Crystals are materials that are made up of a repeating pattern of a simpler "unit cell." Crystal structures are of key interest for many environmental problems, such as materials for solar panels and batteries (Niu et al., 2015).

In contrast to problems in drug discovery, many crystals are not composed solely of organic molecules. Crystals commonly contain many heavy (non-Hydrogen) atoms which make various quantum mechanical calculations of their energetic properties very difficult and time consuming. For example, the common benchmark dataset QM9 for predicting quantum mechanical properties includes over 130,000 molecules contains fewer than 9 "heavy" atoms (Ramakrishnan et al., 2014).

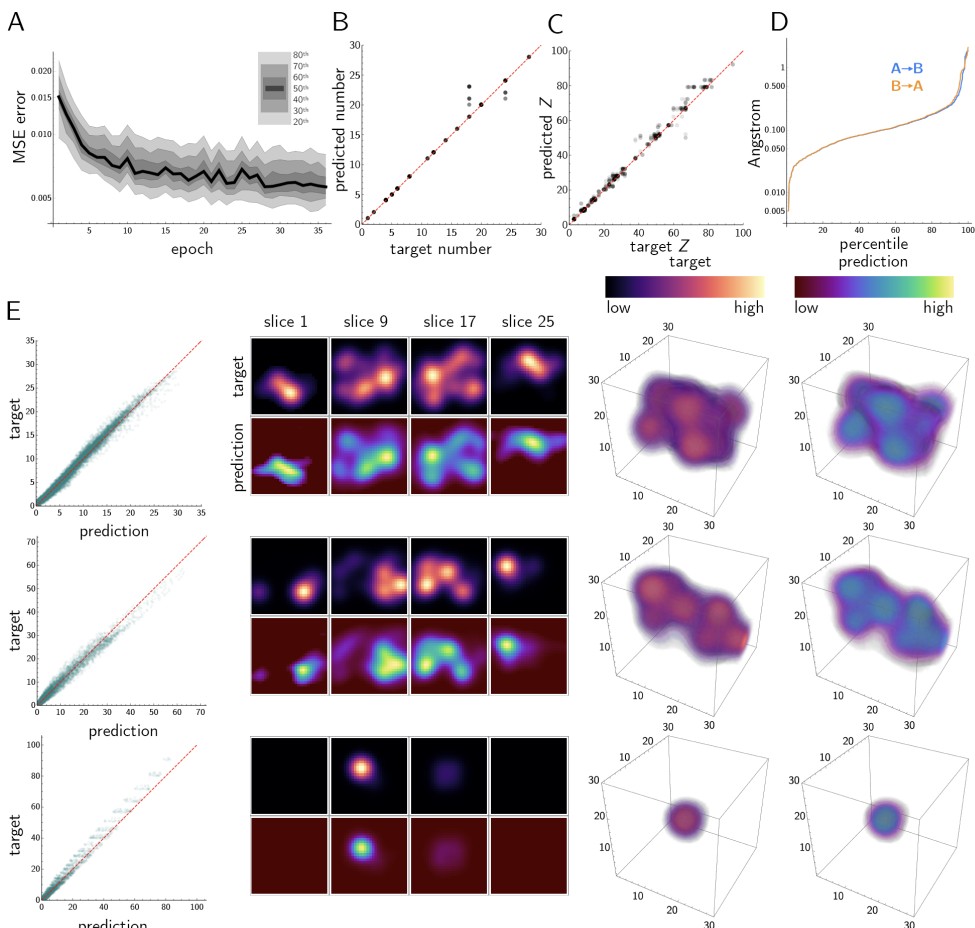

Figure 3: **Single unit cell accuracy**. **(A)** We show the voxel wise reconstruction error (plotted with mean square error) during training. **(B)** For random molecules in the test set, we plot the number of true atoms and the number of recovered atoms after segmentation. **(C)** For the reconstructed atoms, we plot the predicted atomic number versus the atomic number of the nearest true atom. **(D)** We compute the distance from each true atom to the nearest predicted atom and vice-versa (orange and blue, respectively). **(E)** For three different crystals we plot the predicted versus reconstructed density at each voxel. We also show 2-D slices through the target and prediction, along with the 3-D reconstructions. For plotting, the density on each figure is normalized between 0 and 1 though is not decoded as such.

There have been two parallel lines of work applying machine learning to material sciences. The first deals with the prediction of physical properties from a compound without performing a computationally expensive density functional theory (DFT) calculation.

Recently, Xie and Grossman introduced crystal graph convolutional networks (CGCNN) for accurate prediction of 8 different DFT-calculated properties of crystal structures (Xie & Grossman, 2018). MatErials Graph Network (MEGNet) uses graph neural networks (Chen et al., 2019) to predict a variety of energetic properties on both the QM9 dataset (Ramakrishnan et al., 2014) as well 69,000 crystal structures from the Materials Project (Jain et al., 2013). Another network, SchNet, uses 3-D spatial information to directly leverage the interactions between separate atoms. They then use a continuous-filter convolution for accurate property prediction (Schütt et al., 2018). These methods, and many more, have all achieved impressive results on many standard benchmark datasets. Cubuk, *et al.* use transfer learning to search an enormous space of molecules for promising Lithium ion conductors (Cubuk et al., 2019).

Another line of work is concerned with generating molecules, often molecules that have specific properties. In the past decade, generative models for drug discovery have achieved impressive

results. Typically a representation of a compound, such as a `SMILES` string (Gómez-Bombarelli et al., 2018; Segler et al., 2017; Alperstein et al., 2019) or a graph (Jin et al., 2018; Assouel et al., 2018; Mansimov et al., 2019) is encoded and decoded. Then, by sampling the resulting latent space, novel molecules can be generated. Either by performing an auxiliary task with the latent space (as done in Gómez-Bombarelli et al. (2018)) and then performing optimization or by conditioning the latent space on desirable properties, novel chemical structures are obtained.

One possible representation of drug-like molecules is a `SMILES` string. However, a difficulty faced using the `SMILES` representation is ensuring that the decoder decoded a valid `SMILES` string (Gómez-Bombarelli et al., 2018; Kusner et al., 2017). Additionally, there is not a strong notion of distance between molecules and their `SMILES` representation (Jin et al., 2018). One approach, ChemTS, uses `SMILES` strings along with recurrent neural networks and Monte Carlo Tree Search to better ensure generated strings decode to real molecules (Yang et al., 2017). By explicitly penalizing invalid `SMILES` strings in their reward function they are able to bypass the difficulty in their decoded molecules being non-physical.

More recently, graph based methods have proven very successful in generating synthetically attainable molecules. By building a database of molecule fragments (like LEGO bricks), a graph is formed based on the connectivity of different structures (Jin et al., 2018). These methods have far fewer difficulties ensuring that the decoder produces physically valid molecules. For generation of physical molecules, this is a very important consideration, as there are many hard constrains that most be obeyed. Other graph based approaches include flow based models (Madhawa et al., 2019). Also using graph neural networks, E. Manismov and collaborators developed a method to, given a graph, recreate conformations of a 3-D molecule and predict its energetic properties (Mansimov et al., 2019). In Mansimov et al. (2019), the authors propose a graph-based generative method that is able to generate molecules with desired target properties. Impressively, they recover 3-D positional information for the atoms and show they achieve the required accuracy for relaxation.

An additional challenge for generative models in the physical sciences is ensuring that samples from the latent space, $\mathbf{z}$, decode into physically plausible objects. The decoded objects need to obey physical constraints and an object that may appear physical is not certain to actually be experimentally realizable. In many domains this is not of specific concern, for example a generated animal is scored on some likelihood based on its visual appearance, not whether or not the generated creature is genetically possible. Recently, there have been generative models leveraging the 3-D nature of the problem. G-SchNet, a generative model for 3-D molecules (Gebauer et al., 2019) leverages the SchNet architecture- a start-of-the-art property prediction network (Schütt et al., 2018). They show they are able to generate molecules, placing atoms in 3-D space in a rotationally invariant manner. By appropriately conditioning the placement of new molecules based on the location and identity of previous molecules, they ensure the symmetries that are required for the molecules to relax in a DFT calculation. They propose expanding G-SchNet to generating crystal structures as a future direction.

While there is a rich literature in the generation of organic compounds, recently there has been work in the generation of more complex crystal structures, such as CrystalGAN (Nouira et al., 2018). The authors of CrystalGAN introduce geometric constraints and show that their method is able to produce stable structures compared to methods that do not include such domain knowledge such as DiscoGAN (Kim et al., 2017). CrystalGAN uses a dataset with a much larger selection of elements than much previous work, but they limit themselves to molecules of particular chemical structure, nor do they generate a 3-D arrangement. In this work, we do not limit the generated crystals in this way.

## 3 METHODS

We use a dataset containing 46,744 `cif` files which contains the information describing the unit cells of properly relaxed crystal structures from the Materials Project (Jain et al., 2013; Xie & Grossman, 2018). We use 80% of the data for testing and the other 20% for training. Using the Python library `pymatgen` we preprocess all of the data into the density 3-D matrices (Ong et al., 2013) described below. The boundary box of a crystal structure is controlled by six different degrees of freedom (see Fig. 1). Each side can have a different length and the angles between the three sides is also variable. Additionally, the internal complexity of each crystal can also vary widely– some unit cells in our dataset contain a single atom while other unit cells can have over one hundred different atoms. This tremendous variability in structure makes a universal representation difficult.

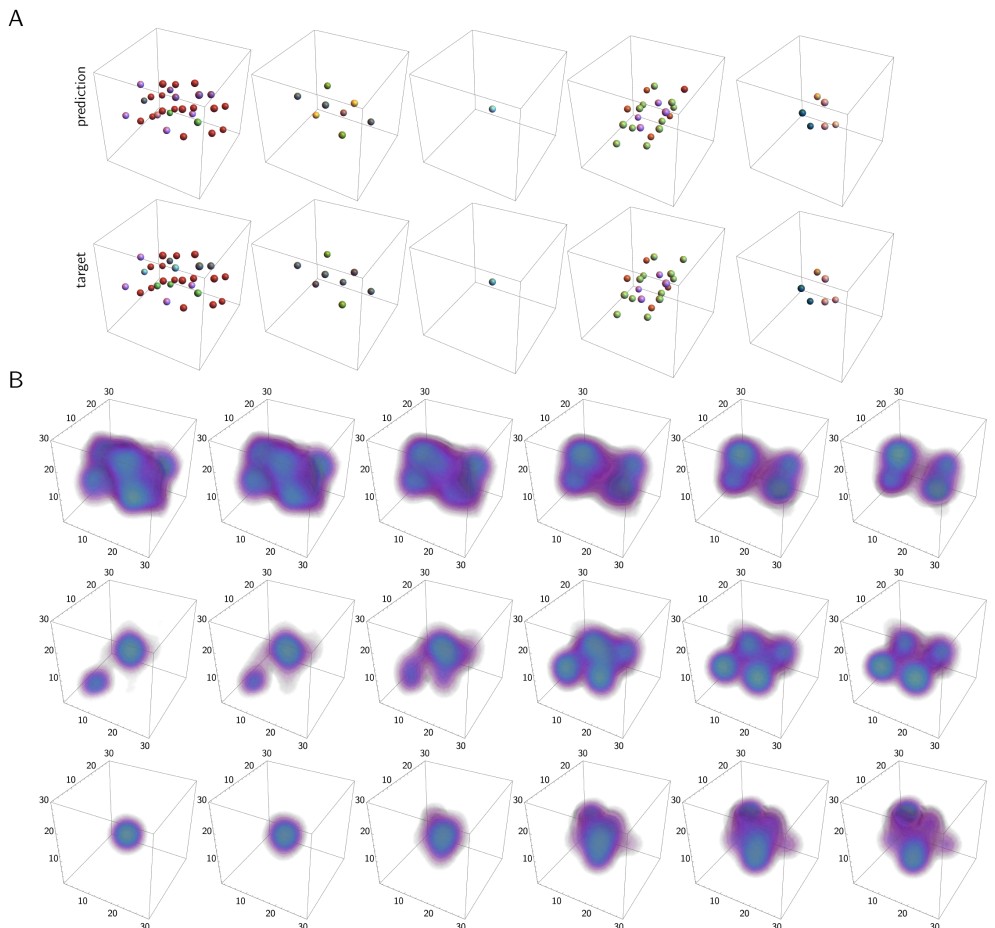

Figure 4: **Species reconstruction and latent space interpolation. (A)** For five different randomly selected crystal unit cells we show the target and our reconstruction. Atoms are colored using default atomic colors, atoms with similar atomic number do not necessarily have similar colors. **(B)** We show the reconstruction of latent space interpolation between two molecules for 3 random sets of targets. For a video, see the Appendix.

## 3.1 Data Representation

As a first step for the problem of generating novel, physical crystal structures we begin with the simpler problem of encoding and decoding physical locations of atoms in space. We begin by considering a cube with side length 10 Å, which we represent by $M$. We divide this cube into 30 equally spaced bins, resulting in a $30 \times 30 \times 30$ cube with each voxel representing 0.33 Å on each side (see Fig. 2 and Fig. 3E). In the appendix, we discuss this choice in more detail. This data representation is similar to many computer vision tasks, so we use similar convolution based network architectures.

We consider the crystal structures in our dataset where the maximum side length is less than 10 Å. We consider two different data representations of different complexity. In the first, we shift each single unit cell to the center of our grid, then we randomly rotate it. In this representation, we encode and decode a single unit cell. We randomly sample 3 different rotations for each crystal. In this case, the encoded structures typically have between 1 and 30 atoms, with a few reaching between 50 and 100. In the second representation, we repeat these unit cells in each direction infinitely and randomly sample a cube in space. This means that each cube contains at least one unit cell, but some will contain more than one cell, since in all cases a cube with 10 Å sides is chosen. We choose one representation where a unit cell begins at $(0, 0, 0)$ and randomly sample 2 other cubes in this space resulting in a dataset of over 100,000 examples. In this representation, we typically encode the

locations of between 20 to 100 atoms, with some having over 200. An advantage of our representation is that we are agnostic to the number of atoms that we are encoding and decoding, something that is difficult for other approaches such as graphs.

Using both these representations, we compute a density field, where each pixel $(i, j, k)$ is defined to have value

$$\boldsymbol{M}_{i,j,k} = \frac{1}{\sigma^3 (2\pi)^{3/2}} \sum_m Z_m \exp\left(-\frac{d(\vec{Z}_m, (i, j, k))^2}{2\sigma^2}\right) \tag{1}$$

Where $d(\cdot, \cdot)$ represents the Euclidean distance between the two arguments. $\vec{Z}_m$ represents the 3-D coordinates of atom $Z_m$. We set $\sigma$ to 1.0 Å. When plotting, we multiply the output by $\sigma^3 (2\pi)^{3/2}$. We do this because when $\sigma = 1$ this results in the value of $Z$ (the atomic number) being present at its location in space. Because of the structure of this representation, when the density of atoms increases, there are more interactions making it harder to reconstruct the true identity of an atom. Ultimately, our aim is to be able to reconstruct the location and species of the different atoms in this grid. To that end, we also construct a species matrix, $\boldsymbol{S}$, where each voxel is either a 0 or equal to the atomic number of an atom that is within 0.5 Å. We construct two neural networks that attempt to learn $\boldsymbol{M}$ and $\boldsymbol{S}$ in parallel. These two networks are trained together and are detailed in the following section.

## 3.2 NETWORK ARCHITECTURE

We use a variational autoencoder (Kingma & Welling, 2013) to encode and decode our 3-D density maps, $\boldsymbol{M}$. We use a $\beta$ multiplier on the Kullback-Leibler (KL) loss term to encourage no correlations between different elements of the latent space (Higgins et al., 2017). Our encoder, $E(\cdot)$, is a convolutional neural network. The decoder, $D(\cdot)$, uses upsampling and convolutions in favor of transposed convolutions (Odena et al., 2016). We use batch normalization (Ioffe & Szegedy, 2015), LeakyReLU activations, and the code is written in Pytorch (Paszke et al., 2017). The optimization is done using the Adam optimizer (Kingma & Ba, 2014). We use a latent space size of 300 for all experiments.

Simultaneously with training the VAE, we train a 3-D U-Net segmentation model, $U_{net}(\cdot)$, with an attention mechanism to segment the output of the decoder (Ronneberger et al., 2015; Oktay et al., 2018). For accurate segmentation, especially in the case of a repeating lattice, it is important to be able to capture the dependencies between different atoms. We found that a U-Net model worked well, though future work will explore sequentially classifying atoms. Ideally, the primary work is done by the VAE and the role of U-Net is to perform the tedious optimization problem of recovering the species identity and locations that is most consistent with the reconstructed density field.

We include a weighted (by $\gamma$) loss from the segmentation in the loss of the encoder/decoder. In our experiments, we set $\gamma$ to 0.1. We experimented with $\gamma = 0$ and $\gamma = 0.33$ and found that 0.1 proved an acceptable intermediate. With $\gamma = 0.33$ we found slightly improved segmentation results but worse density reconstruction results. When we completely removed the effect of the species loss from the density reconstruction, species reconstruction results degraded.This allows us to train the entire network in an end-to-end fashion. The entire model is shown in Fig. 2.

We use three terms in the loss function for the VAE with the most weight given to the reconstruction of the density matrices. The segmentation network is only concerned with the final segmentation of the reconstruction from the decoder. Currently, we treat each atom type as a different class, with an additional (most common) class for "no atom." The loss function used for the VAE is

$$\mathcal{L}_{\text{VAE}} = L_{\text{RE}}(\hat{\boldsymbol{M}}, \boldsymbol{M}) + \beta(D_{\text{KL}}(q(\mathbf{z}|\boldsymbol{M})||p(\mathbf{z}))) + \gamma L_{\text{BCE}}(\hat{\boldsymbol{S}}, \boldsymbol{S}). \tag{2}$$

The first term is the reconstruction error and the third term is the binary cross entropy loss from the segmentation. The second term is the Kullback-Leibler divergence between the prior (set to $p(\mathbf{z}) = \mathcal{N}(0, 1)$) and $q(\mathbf{z}|\boldsymbol{M})$ the approximate posterior of the latent vector given input $\boldsymbol{M}$. We use a loss given by $\mathcal{L}_{\text{U-Net}} = L_{\text{BCE}}(\hat{\boldsymbol{S}}, \boldsymbol{S})$ for the segmentation. In the above expression, $\boldsymbol{M}$ is the input density field. $\mathbf{z} = E(\boldsymbol{M})$ and $\hat{\boldsymbol{M}} = D(\mathbf{z})$ is the reconstructed density field. The variable $\boldsymbol{S}$ represents a 1-hot species matrix. Lastly, $\hat{\boldsymbol{S}} = U_{net}\left(\hat{\boldsymbol{M}}\right)$ represents the probability matrix from the segmentation routine, which will be of shape: [batch $\times$ classes $\times$ x-dim $\times$ y-dim $\times$ z-dim].

## 4 RESULTS

We find that we are able to accurately encode and decode 3-D representations of density fields (see Figs. 3 and S8). Using the segmentation network, we segment the output of the decoder into distinct atoms. Using a trained network, we show that random samples from the latent space decode to samples that obey many of the same statistics as the training distribution (see Fig. 5 and Fig. S7). Additionally, in the case of a repeating unit cell, we alter our training routine to condition the generation of crystals on the largest atomic number present.

### 4.1 PLACEMENT OF ATOMS

Using the output of the segmentation network, we take the $\tilde{\mathbf{S}} = \text{argmax}(\hat{\mathbf{S}})$. From this representation, we find the connected components and use majority voting to assign each cluster an atom identity (see Appendix for details). In Fig. 3 (and Fig.S8) we show the accuracy of all top-1 predictions. We compare the results of our segmented matrix, $\tilde{\mathbf{S}}$ with the true values, $\mathbf{S}$. From the center of mass of each predicted atom $i$, we compute the distance to the nearest true atom as follows

$$\min_j d(\tilde{\mathbf{S}}_i, \mathbf{S}_j) \, \forall \, j \text{ and } \min_i d(\tilde{\mathbf{S}}_i, \mathbf{S}_j) \, \forall \, i. \quad (3)$$

Similarly, we can compute the distance from each true atom to the nearest predicted atom. By computing this metric in both directions, we are able to verify that our model is placing atoms in the correct locations but only in those locations.

Table 1: Location and species reconstruction error. $P$ denotes prediction and $T$ denotes target. Distances in angstrom. For $Z$ reconstruction, we show the percentile at each absolute difference.

| Locations | | |
|---|---|---|
| Percentile | $P \to T$ (Å) | $T \to P$ (Å) |
| 0. | $4.31 \times 10^{-4}$ | $5.47 \times 10^{-4}$ |
| 10. | $5.03 \times 10^{-2}$ | $5.02 \times 10^{-2}$ |
| 20. | $7.04 \times 10^{-2}$ | $7.05 \times 10^{-2}$ |
| 30. | $9.12 \times 10^{-2}$ | $9.13 \times 10^{-2}$ |
| 40. | $1.14 \times 10^{-1}$ | $1.14 \times 10^{-1}$ |
| 50. | $1.36 \times 10^{-1}$ | $1.37 \times 10^{-1}$ |
| 60. | $1.60 \times 10^{-1}$ | $1.61 \times 10^{-1}$ |
| 70. | $1.94 \times 10^{-1}$ | $1.94 \times 10^{-1}$ |
| 80. | $2.48 \times 10^{-1}$ | $2.50 \times 10^{-1}$ |
| 90. | $3.94 \times 10^{-1}$ | $4.03 \times 10^{-1}$ |
| 100. | 2.59 | 1.89 |
| **Species** | | |
| Abs. Diff. in $Z$ | Percent | |
| 0 | 66.8 % | |
| 1 | 25.1 % | |
| 2 | 5.8 % | |
| > 3 | 2.2 % | |

### 4.2 ACCURACY ON A UNIT CELL

In Fig. 3 we plot the results of applying our model to single unit cells of crystals (see also Table 1). We find out model is able to very accurately segment the locations of atoms with nearly 99% of atoms placed within 0.5 Å of their true location. Nearly 90% of unit cells are reconstructed with the correct number of atoms and 97% of unit cells are reconstructed with less than 2 atoms extra or missing. 66% of species are correctly classified in the test set, with nearly 100 possible classes. When an atom is incorrectly classified, it is usually by one or two atomic numbers (see Table 1). Often adjacent elements (with atomic numbers that differ by 1) have very different chemical properties. We retrained the network to predict atomic group (where elements in the same group tend to have more similar properties) and found 93% accuracy. We report accuracy on $Z$ as this is a more difficult problem and in both cases we are still unable to reconstruct crystals with the accuracy to relax in a DFT calculation. Missed atoms typically occur near the boundary.

Next, we will discuss the same network applied to repeating unit cells. In all cases, the results for single unit cells are improvements over the results of multiple unit cells. For most figures, there are corresponding panels between the single unit cell and repeated cell results.

### 4.3 ACCURACY ON REPEATING UNIT CELLS

In Fig. S8C we show the percentiles of all pairwise minimum distances in our reconstructed samples. We find that 50% of all reconstructed atoms are in 0.2 Å. For both directions, the 75th percentile error was under 0.25 Å and the 90th percentile of reconstructed atoms were within 0.5 Å. For predicted atoms that are within 0.5 Å of a true atom's location, we find that 65.4% of the time our prediction matches the atomic number exactly Fig. S8E. Again, nearly all errors occur near the boundary.

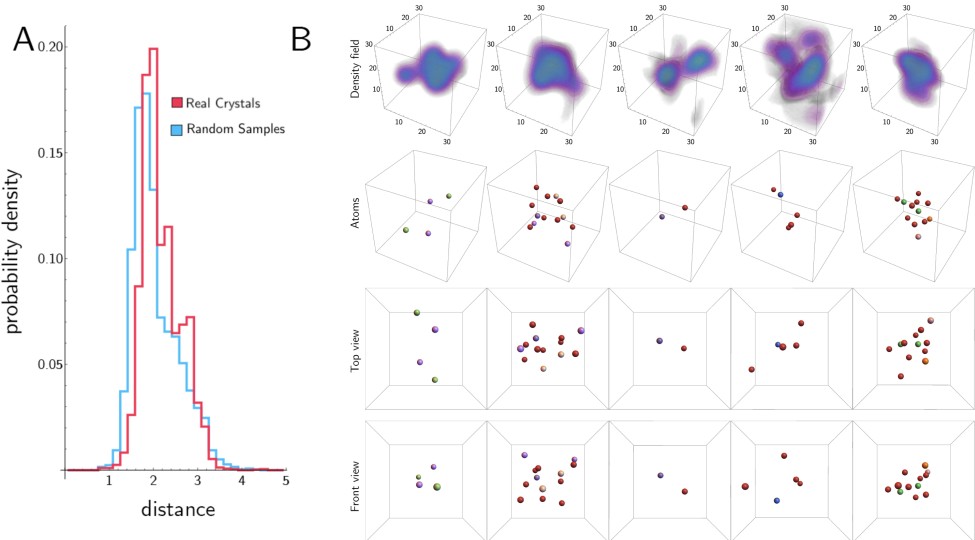

Figure 5: **Decode random latent space vectors**. (**A**) We look at the spacing between nearest atoms from random draws from the latent space compared to those from real crystal structures. In blue we show the distribution of random reconstructions. In red we show the distribution of true inter-atomic spacing. (**B**) For 5 different random latent space vectors, we show the reconstructed density field and the resulting segmentation. Notice that in some cases small amounts of density are predicted but are not segmented into an atom (for example, in columns 3 and 4).

For 120 randomly selected reconstructed crystals from the test set, we compare the number of atoms after segmentation with the true number (Fig. S8D and Fig. 3B). In Fig. S8E we compare the predicted species compared with the species of the nearest true atom, independent of the distance to the nearest atom. We find that based on the results of Fig. S8, our current model is able to more accurately reconstruct the locations of atoms than obtain their true atomic number. This makes sense, given the enormous number of possible classes. However for repeating unit cells, when an atom is correctly predicted to be within 0.33 Å of a true atom location nearly 70% of the time the atom is assigned exactly the correct $Z$ value. The correlation between the true and predicted value is 0.98 suggesting that when the network makes an incorrect assessment, it chooses a nearby atomic number.

Further improvements in this placement and identification of atoms is necessary in achieving crystals that will appropriately relax in a DFT simulation for the calculation of quantum mechanical properties. However, a very exciting immediate application of this work is in random structure searching Pickard & Needs (2011).

### 4.4 LATENT SPACE INTERPOLATION

We encode two different true density maps into vectors $\mathbf{z}_1$ and $\mathbf{z}_2$, respectively. We then linearly interpolate between these two values in latent space, constructing intermediate vectors $\tilde{\mathbf{z}}_i$. We decode the resulting latent space vectors which result in predicted density maps, $D(\tilde{\mathbf{z}})$. Passing these into the trained segmentation routine, we find that intermediate results segment into atoms (see results on a single unit cell in Fig. 3 and Fig. 4). See results on a repeating lattice in Figs. S9 and S11)[1].

### 4.5 RANDOM DRAWS

We can randomly sample a latent space vector $\mathbf{z} \sim \mathcal{N}(0, 1)$ and we then decode the output. We do this for both single unit cells as well as for repeating lattices (see Appendix). Random structure searching is an important area of material science research an exciting direction for this work Pickard & Needs (2011). Using the network trained to encode and decode single unit cells, we decode a $\mathbf{z} \sim \mathcal{N}(0, 1)$.

---

[1]See anonymized link: `https://sites.google.com/view/encode-decode-3d-crystals/home`

In Fig. 5 we show the decoded density fields along with the resulting segmented output. There are a few desirable features: single cells are centered, atoms tend to be of similar atomic number, even in different parts of the reconstructed output, and atoms are not ever too close together.

## 5 CONCLUSIONS AND FUTURE WORK

The ability to encode and decode 3-D structures is a very interesting direction of recent work and increasingly important in helping create models that can understand the world that we live in. In current material design, a lot of focus has been on molecules where the 3-D structure can be safely ignored. However, in many cases this is not the case. We do not know of an effective data representation for these 3-D molecules and find that by directly utilizing a proxy for density we are able to encode and decode the geometry of these molecules.

An important future direction of research is to come up with successful representations for encoding crystal structures such that they can be easily encoded and decoded. Coming up with a representation of crystal unit cells (and 3-D structures in general) so that decoded molecules are physically plausible is an important aspect that needs to be carefully considered.

Our approach is currently unable to generate molecules that are physically stable, but there are promising directions in this direction. Using ideas similar to G-SchNet, using the decoded density field, atoms can be placed sequentially, conditioned on the placement of previous atoms (Gebauer et al., 2019). Working towards decoding molecules that are able to relax is a very exciting direction.

Another improvement would be to alter the structure of our encoder/decoder. Currently, we use standard 3-D convolutions. However, using SE(3) equivariant kernels from M. Weiler *et al.* is a promising direction for improved performance (Weiler et al., 2018). We feel that our current approach could be extended beyond the domain of material science, as we are able to encode a 3-D distance map, this approach could be broadly applied to many objects. In essence, we are able to learn a distance transform from an object. This is more general than for encoding and decoding atomic structures. To try to facilitate further research in the creation of 3-D representation of more complex atomic structures, we will release the code shortly.

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

## A    Supplemental Information

### A.1    Videos

See anonymoized project site at: `https://sites.google.com/view/encode-decode-3d-crystals/home`.

## B    Choice of Data Representation

We choose to use a 10 Angstrom cutoff as it kept over 90% of the data we had available without becoming too computationally demanding, as we needed to maintain a relatively small grid spacing between voxels (0.33 Angstrom). Because the problem is 3-D, a $30^3$ input is equivalent to a $164 \times 164$ image, in terms of pixels.

## C    Comparison of 1-D, 2-D, and 3-D Representations

For small molecules, 1-D `SMILES` and 2-D graph based representations have been very effectively used for property prediction and for molecule generation tasks. In a `SMILES` representation, a specific grammar is learned but from a valid string, a specific molecule is generated based on the bond arrangement Weininger (1988); Gómez-Bombarelli et al. (2018). Similarly, a graph based representation is able to place atoms in space without specifically assigning each atom a specific coordinate Jin et al. (2018). Additionally, graph representations have been shown to be successful for property prediction of crystals Xie & Grossman (2018), but not for generating crystals. For small molecules, these representations have proved very effective, especially since both these representations effectively reduce the dimension of the problem.

Specifically, `SMILES` and graph representations both **do not encode the specific spatial arrangement of atoms**. Instead, these representations are cleverly created in such a way that the spatial arrangements can reconstructed. Unfortunately, for many larger molecules these representations are not sufficient. Therefore, we focus on developing a representation that is able to directly reconstruct both species and locations in a 3-D space.

## D    Species Prediction

Each voxel is assigned an identity based on the `argmax` of the U-Net. The background has label 0. First, we find all connected components of non-background voxels. For each connected component, we tabulate the species predicted at each voxel and each cluster is assigned an identity baased on the majority vote. The location is based on the centroid of the cluster. We experimented with more complicated procedures (such as weighting by the probability vector) and did not find a meaningful improvement.

## E    Training Details

We used a Tesla V100 for training. We used a learning rate of 1e-5 and a batch size of 24.

### E.1    Network Details

#### E.1.1    Encoder

3D convolution, kernel size of 5 and stride of 2. 16 channels.
Batch Norm and LeakyReLU activation.
3D convolution, kernel size of 3 and stride of 1. 32 channels.
Batch Norm and LeakyReLU activation.
3D convolution, kernel size of 3 and stride of 1. 64 channels.
Batch Norm and LeakyReLU activation.
3D convolution, kernel size of 3 and stride of 2. 128 channels.

Batch Norm and LeakyReLU activation.
Fully connected layer with a bottleneck size 300.

### E.1.2 DECODER

Fully connected layer.
Reshape to 128 5,5,5
Trlinear upsample by a factor of two.
Conv3D with 64 channels, kernel size of 5, leakyReLU.
Trlinear upsample by a factor of two.
Conv3D with 32 channels, kernel size of 5, leakyReLU.
Trlinear upsample by a factor of two.
Conv3D with 16 channels, kernel size of 4, leakyReLU.
Conv3D with 1 channels, kernel size of 4, ReLU.

## F RESULTS FROM THE ENCODER-DECODER

Our encoder-decoder architecture is able to accurately reconstruct the voxel-wise density (Fig. S8A and Fig. S6). We find a strong correlation between the predicted and target voxel value (Fig. S6A). This is essential for achieving an accurate segmentation of the correct atoms and their corresponding positions. Due to the strong penalty on the KL term we find that there is an increased spread in the resulting predictions when attempting to encode and decode repeating unit cells. This is less of an issue when encoding and decoding single unit cells.

## G ACCURACY OF RANDOM DRAWS IN A REPEATING LATTICE

To test whether random samples from the latent space, $\tilde{\mathbf{z}}$, decode to physically realistic molecules, we trained a discriminator. We trained an auto-encoder that was able to very accurately reconstruct molecules by greatly reducing the penalty on $\beta$ (see Fig. S12). Then, we compute real latent space representations of our different molecules, $\mathbf{z}$. We then draw a $\lambda \sim U(0,1)$ and construct a new latent space vector

$$\hat{\mathbf{z}} = \lambda \tilde{\mathbf{z}}_{\mathcal{N}(0,1)} + (1-\lambda)\mathbf{z}_{\text{real}} \tag{4}$$

where $\tilde{\mathbf{z}}$ is a random sample from a unit normal and $\mathbf{z}_{\text{real}}$ is a random "true" latent space. We pass this through the decoder and provide this a label $(1-\lambda, \lambda)$. We then randomly draw $\mathbf{z} \sim \mathcal{N}(0,1)$ from our trained network and pass this reconstruction $\tilde{\boldsymbol{D}}$ into the discriminator network, which outputs a prediction of the distance from a true crystal reconstruction versus a random draw from a latent space of a previously trained network. Applying this network to random samples from $\mathbf{z} \sim \mathcal{N}(0,1)$ we get a mean value of 0.84 with a standard deviation of 0.05. For true decoded samples, we find a mean value of 0.90 with a standard deviation of 0.01. In Fig. S13, we show results of the trained network and predictions on three different datasets. We find that the target latent space decodes to high-scoring samples (see Fig. S13B) and that out of distribution samples typically decode to much lower scores.

### G.1 CONDITIONAL VAE

Moving forward, an important step will be creating structures that have specific chemical properties. Moving towards this direction, we attempt to condition the generation of molecules on the largest species present (Sohn et al., 2015). To do this, we multiply the input and output of the bottleneck by the largest present density. By doing this, we are still able to get the network to train but find that we are able to control the magnitude of the resulting density field without perturbing the geometry (see Fig. S10B). We attempted to condition the encoder and decoder by concatenation though we found this was not sufficient to generate density fields with the desired property.

The ability to condition on specific properties has obvious applications in the targeted generation of molecules. In this case, we choose to condition on the largest present density as a way to generate molecules that do not have an atom with more than a specified maximum atomic number present. We

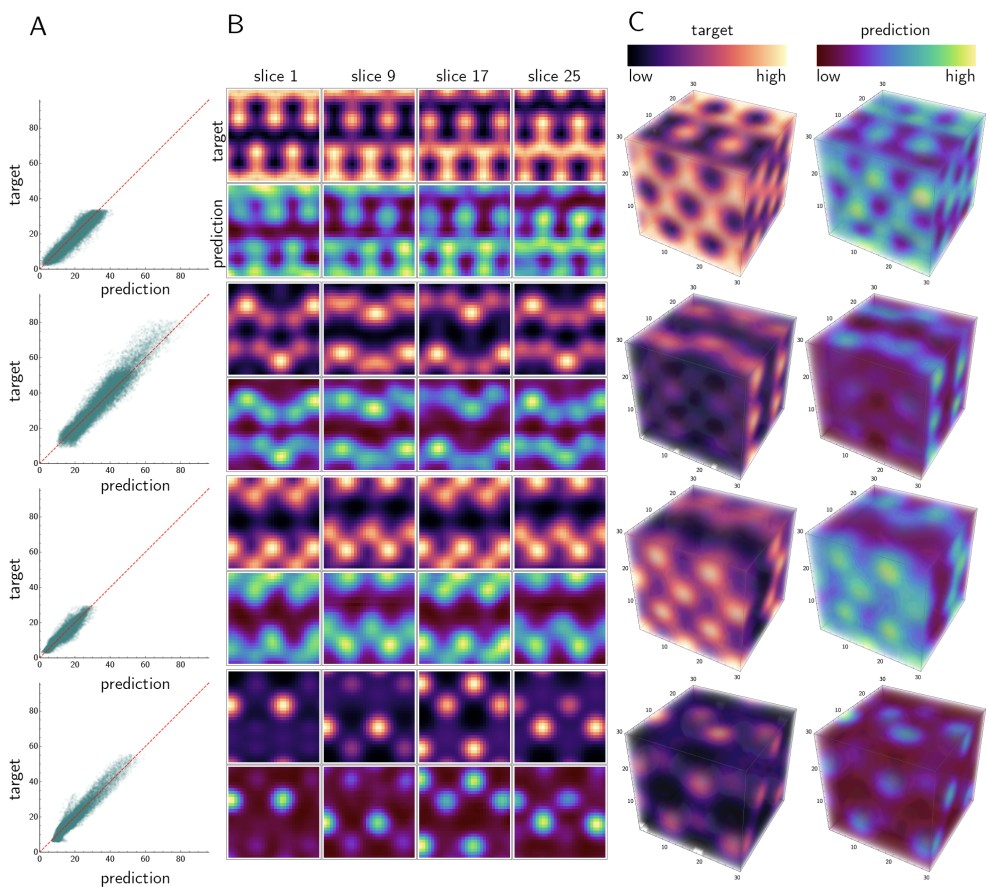

Figure S6: **Accuracy of Model**. **(A)** For each position in 3-D space, we plot the predicted and target density for 4 different random crystals from the test set. The red dashed line is an identity line. **(B)** For each of the panels in **(A)**, we show 4 different $z$-slices through the true and predicted density fields. **(C)** We show the full 3-D reconstruction of the prediction and the true density field.

find that by varying different parameters in the latent space, some vary the field in such a way that appears to correlate with the location of atoms (see Fig. S10C,D). However, unlike as in work on 2-D problems, the changes in the 3-D scalar field are much harder to interpret (Chen et al., 2016; Higgins et al., 2017). Future work will attempt to use improved factorizing techniques (Kim & Mnih, 2018; Chen et al.). Additionally, one could seek to condition the generation of single unit cells on quantum mechanical properties or by adding an auxiliary loss using the latent space to predict quantum mechanical properties.

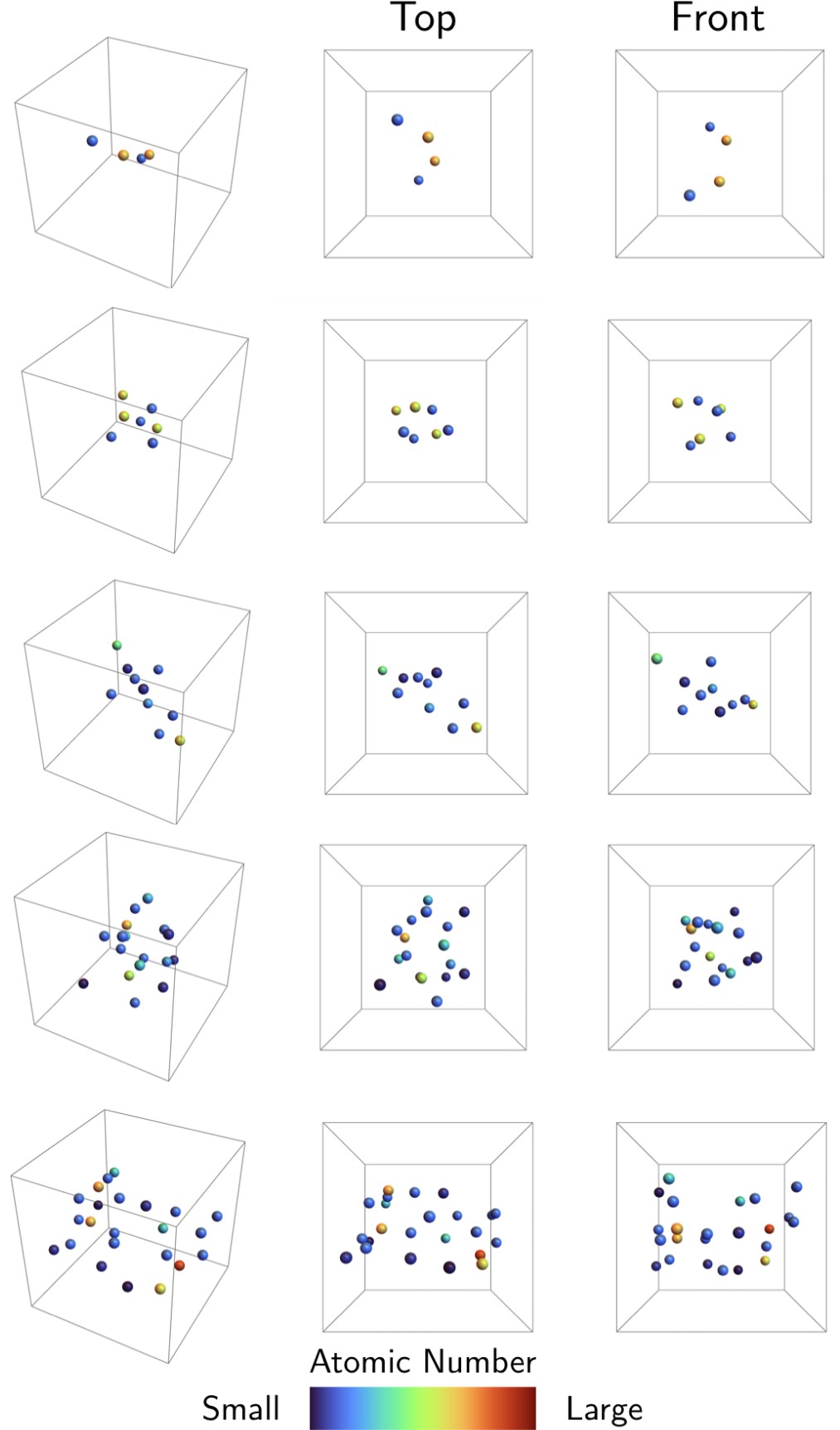

Figure S7: **Random Decoded Samples**. We randomly sample the latent space $z \sim N(0, 1)$. We show a series of examples (from three different views) of different complexity. Note that while symmetry is not preserved, various geometric motifs are.

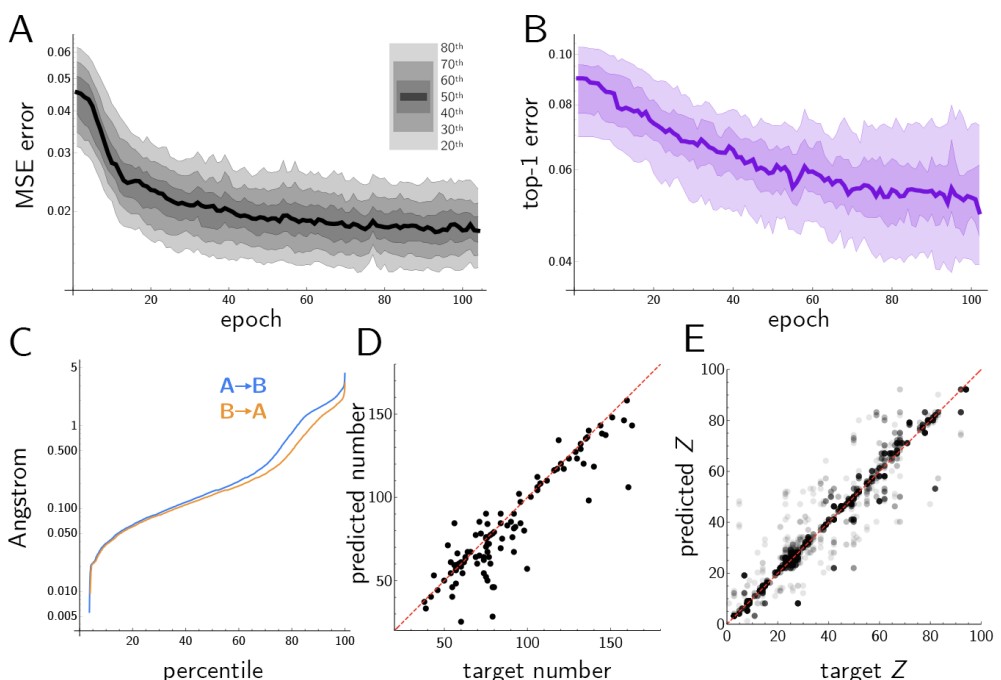

Figure S8: **Repeating unit cell accuracy**. **(A)** For each position in 3-D space, we compute the difference between the truth and the reconstruction. We show different percentile bands of the reconstruction error between the target and predicted density, plotted using the mean square error (MSE). **(B)** For the species matrix from U-Net we ask what the error of top-1 predictions is between our predictions and the ground truth. **(C)** For our segmented matrices, we ask the distance from the nearest segmented atom to the ground truth. We show the distance errors by percentile for both the nearest true atom to each predicted atom and vice-versa (orange and blue, respectively). **(D)** After segmenting the reconstructed density maps we show the predicted and true number of atoms. **(E)** We plot the predicted nearest atom species versus the true species of the closest corresponding atom, as long as the distance is less than 0.5 Å. We find 65.4% are correctly predicted.

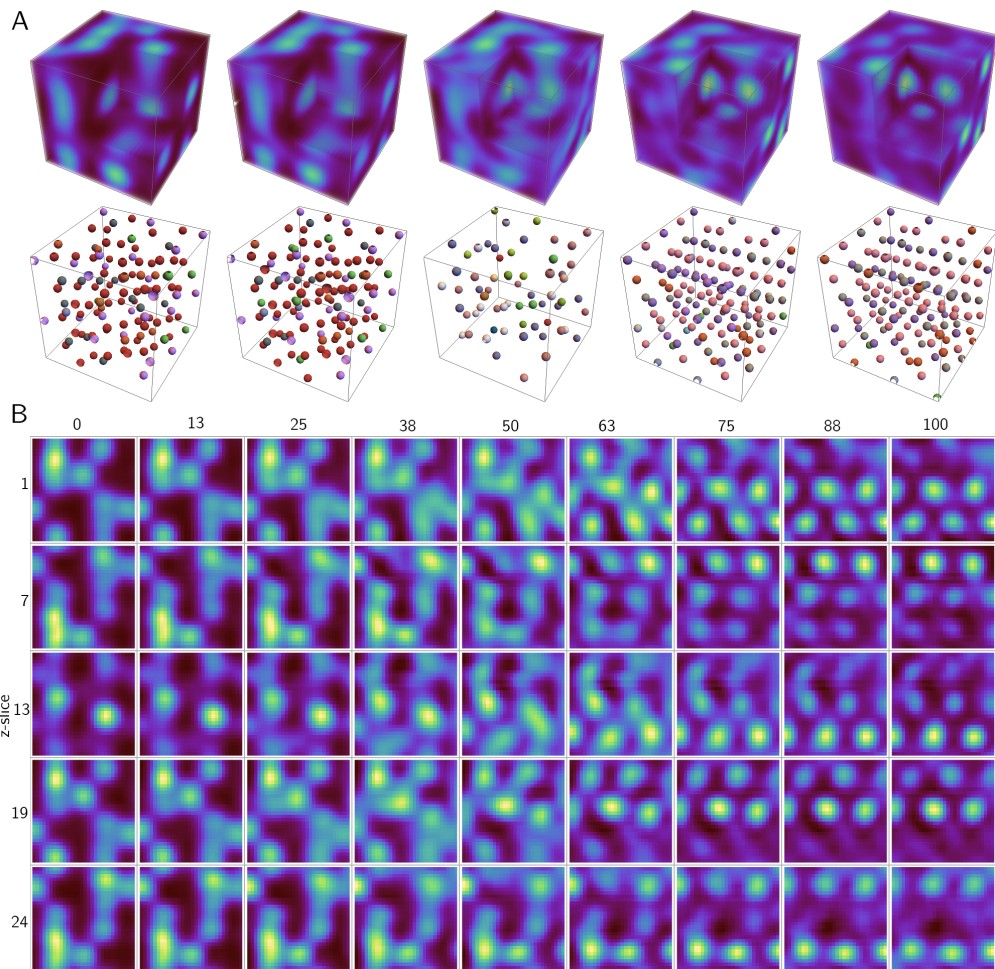

Figure S9: **Interpolation between two molecules**. In **(A)** we show the interpolation between two crystal density maps. We show three equally spaced intermediates the corresponding segmentation. In **(B)**, we show the same interpolation but highlight different two dimensional slices (y-axis). Along the x-axis we label the fraction of the way between the two latent space vectors.

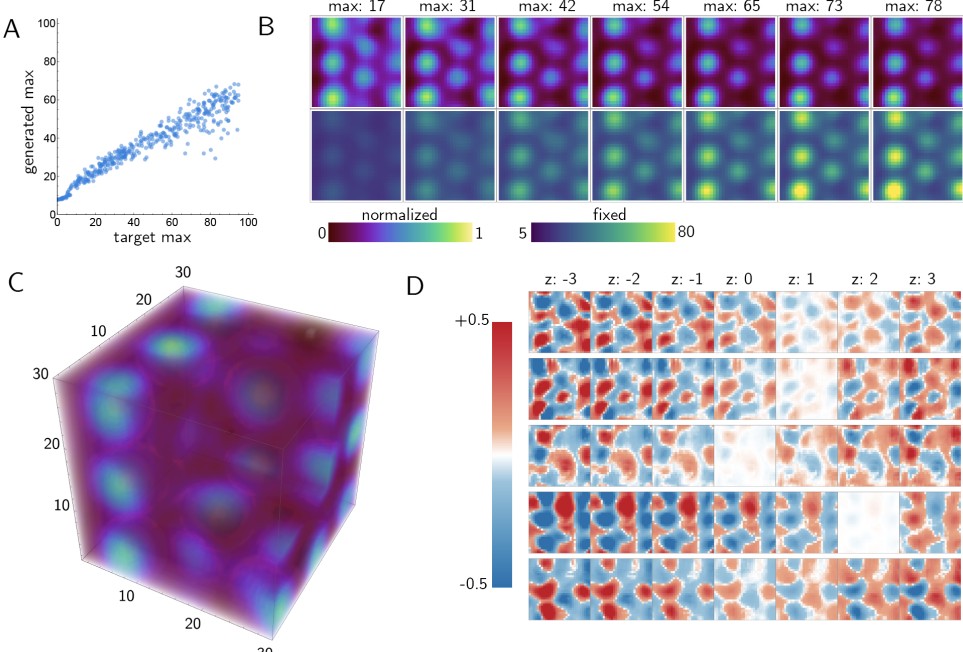

Figure S10: **Conditional generation of molecules**. We multiply the input and output of the bottleneck by the maximum density during training. Then, randomly decoding samples of $\alpha\mathbf{z}$ where $\mathbf{z} \sim \mathcal{N}(0, 1)$ and $\alpha$ is a random target, we find we are able to generate density maps that decode appropriately. If we take a true decoded value and re-scale the output we find we are able to control scale without affecting geometry. **(A)** We show the target and generated max value. **(B)** Using an encoded $\mathbf{z}$, we multiply it by different target values and decode them. We find that we are able to change scale without affecting the geometry. **(C)** We show a decoded sample. We cut away the right most corner facing the viewer. **(D)** For the molecule shown in **(C)**, we show the results of varying one value in the latent space from -3 to 3. We plot the difference in the slice when compared the unaltered, decoded, $\mathbf{z}$ for a slice that is in-plane with many atoms.

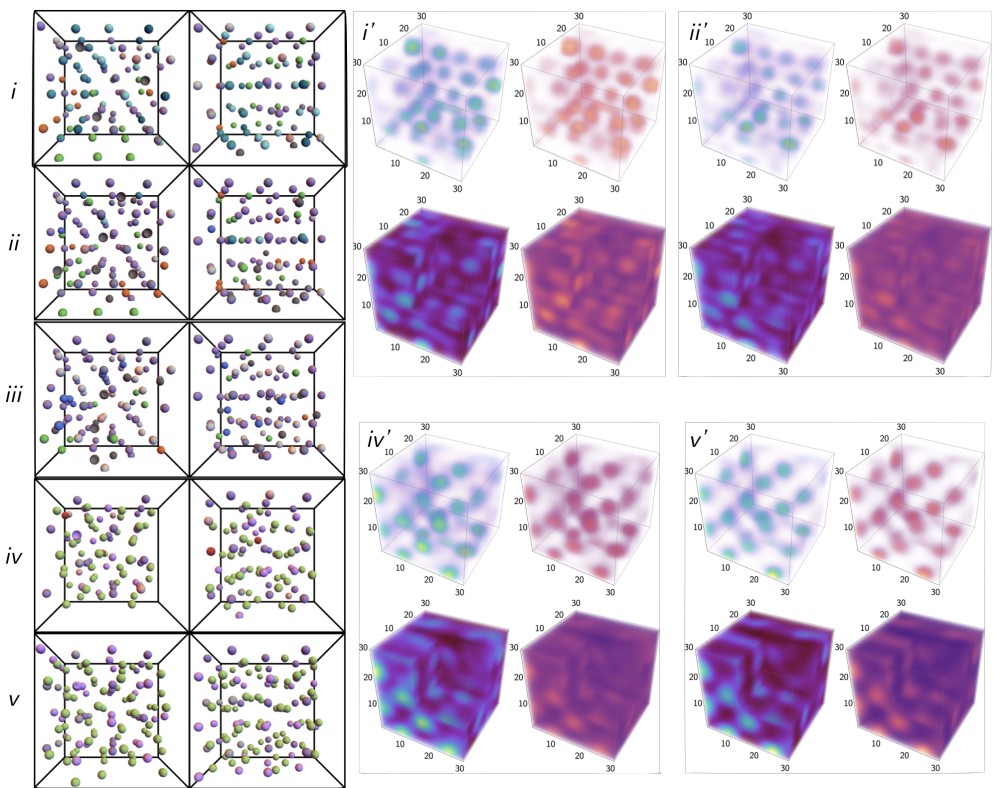

Figure S11: **Interpolation between two molecules**. From $i - v$ we show two views of the output from the segmentation routine decoding the latent space between two different molecules. In the primed corresponding figures we show the density fields that are output by the decoder. We show the output of each grid normalized between 0-1 in blue-green and on a fixed color map in magma. We also vary the opacity to show the locations of predicted atoms.

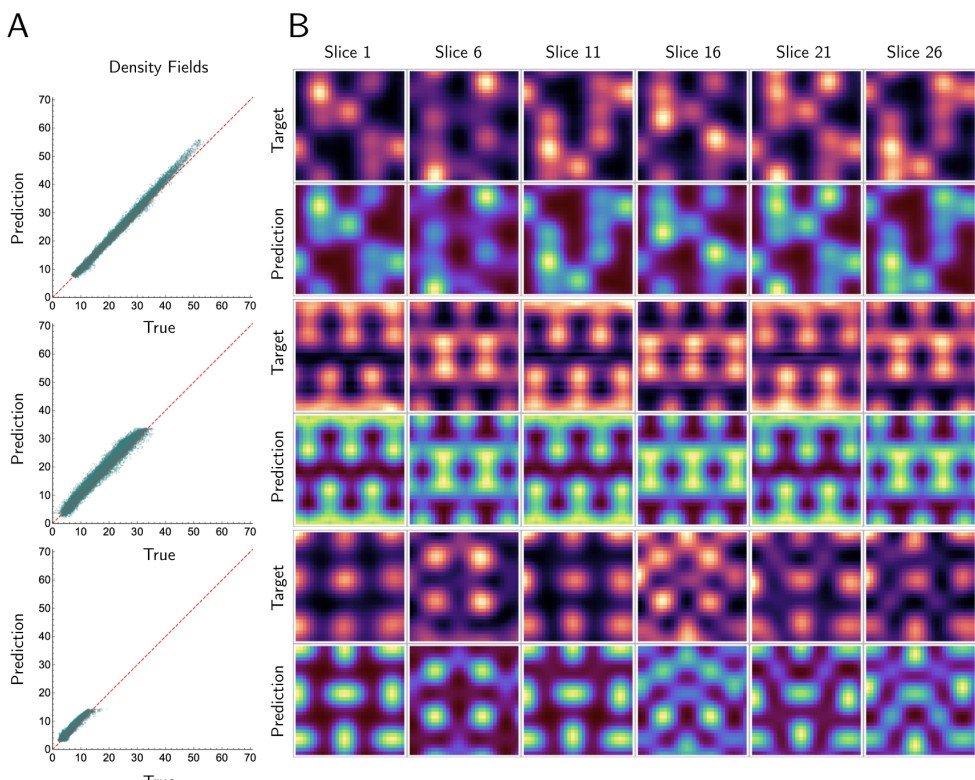

Figure S12: **Results with smaller** $\beta$. We decrease $\beta$ by a factor of 10. By reducing the penalty on the Kullback-Leibler (KL) term we are able to generate more accurate reconstructions. However, sampling $\mathbf{z} \sim \mathcal{N}(0, 1)$ does not fully sample the encoded space of molecules.

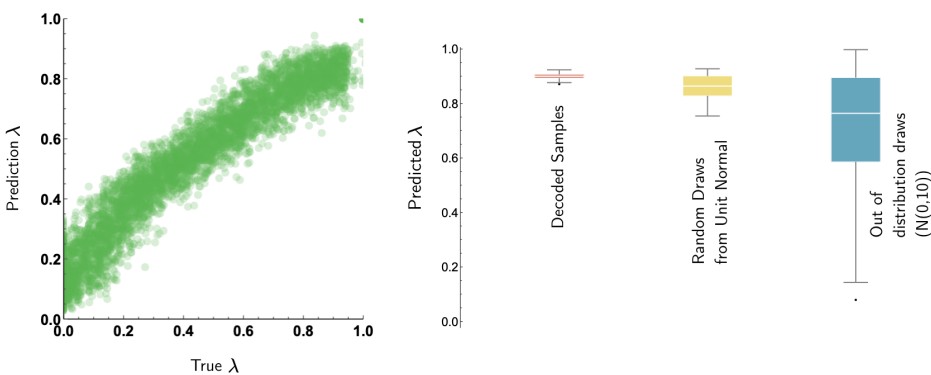

Figure S13: $\lambda$ **prediction for different samples**. Using the network in Fig. S12 we train a discriminator network to predict the "distance" from a real crystal. We apply this discriminator to real decoded samples, random latent space draws, and out of distribution draws (right).

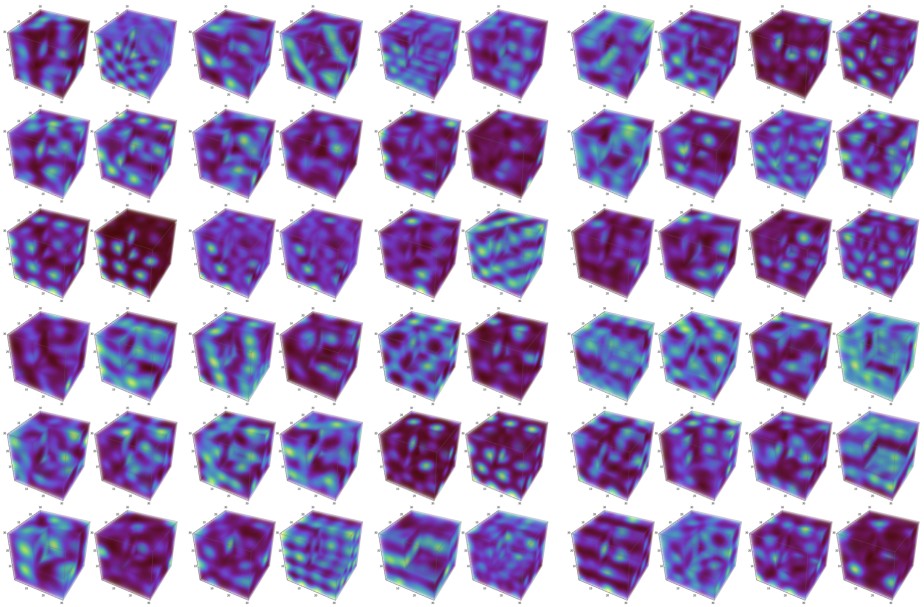

Figure S14: **Decoded samples of repeating unit cells**. Results from the encoder-decoder.

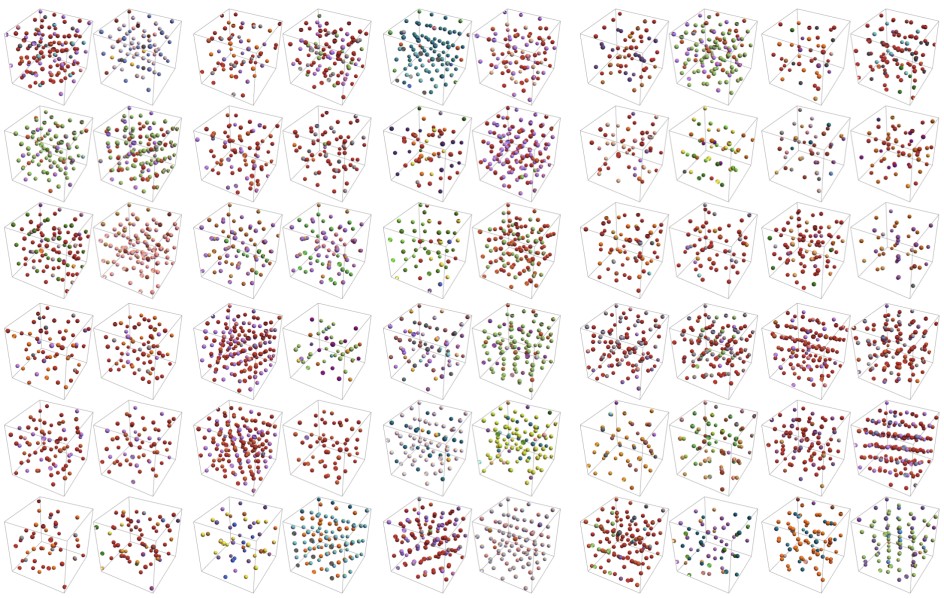

Figure S15: **Segmented Output from the decoded samples**. The segmented output from Fig. S14.

