# OpenReview forum: "Data-Driven Approach to Encoding and Decoding 3-D Crystal Structures"
_ICLR.cc/2020/Conference — Reject_

### Official Review · AnonReviewer2 · 2019-10-23
**Official Blind Review #2**

**Rating:** 8

**Review:**

The paper deals with accurately encoding and decoding 3D atomic positions and the crystal’s species using 2 sets of neural networks a) a VAE that builds a compressed latent space representation of a crystal and b) a UNET for segmenting the latent space into atoms and assigns each atom to its atomic number. Experiments were conducted on over 120K 3D samples of crystals and the results seem to be promising.

The paper is neatly written and well organized.

Comments:

A) Figure 2: For completion, consider marking M and S as outputs of the VAE and U-Net respectively.

B) In Section 3.1, why do the authors use a cube with side of ’10’ Angstrom? And why divide the cube into ’30’ bins?

C) The paper revolves on vanilla 3D convolutions of the crystal structures. Have the authors considered how the results would change if SO(3) rotation invariant convolutions were used instead. The SO(3) convolutions would empower capturing all possible rotations of the crystal other than only its canonical form.

**Experience Assessment:**

I have read many papers in this area.

**Review Assessment: Checking Correctness Of Derivations And Theory:**

N/A

**Review Assessment: Checking Correctness Of Experiments:**

I carefully checked the experiments.

**Review Assessment: Thoroughness In Paper Reading:**

I read the paper at least twice and used my best judgement in assessing the paper.

---

> ### Author Response · Authors · 2019-11-08
> **Thank you for your feedback**
>
> Thank you for your feedback on our work and useful comments.
>
> Thank for for the suggestion  on Figure 2. We will update this in the draft and upload shortly.
>
> >> In Section 3.1, why do the authors use a cube with side of ’10’ Angstrom? And why divide the cube into ’30’ bins?
> This is a useful point, and we should be more clear in the text. We played with a variety of sizes for both values, and  found that this was a balance between (a)  including as much data as possible, (b) having grid sizes that enabled reasonable compute time under our conditions and (c ) was meaningfully accurate.
>
> >> The paper revolves on vanilla 3D convolutions of the crystal structures. Have the authors considered how the results would change if SO(3) rotation invariant convolutions were used instead. The SO(3) convolutions would empower capturing all possible rotations of the crystal other than only its canonical form.
> This is a very useful question, and something  we are actively working on. However, we generally found that the encoder-decoder accuracy is quite good and the weakness is in the difficult segmentation task.

---

### Official Review · AnonReviewer3 · 2019-10-26
**Official Blind Review #3**

**Rating:** 1

**Review:**

PAPER SUMMARY: This paper addresses the problem of encoding and decoding 3D chemical structures, with the ultimate goal of generating 3D crystal structures. The authors propose an auto-encoder framework for encoding the 3D locations of atoms in the crystal to a latent representation and then decoding that representation back into 3D structure. The paper's contributions can be summarized as follows:
1) A data representation that converts the 3D atom locations to a 3D voxel density map, so that they can be encoded by a standard 3D convolutional network.
2) A decoder network that first estimates a 3D density map from the latent vector using upsampling and convolutions and then classifies the atom type (atomic number) per voxel using a 3D segmentation.
3) The network is applied on unit cells of crystals or repeated unit cells from a dataset of crystal structures and the network is shown to be able to accurately reconstruct the 3D density maps, predict the number of atoms in the cell and perform fairly well in their classification into atomic types.

I appreciate that the paper addresses an interesting problem that is not sufficiently explored and can motivate the development of novel methods that generate 3D molecules with particular structure and multiple types of atoms, however the current work combines existing methods, without any architectural modifications that exploit the new domain.  If the presentation of results and experiments is improved, this could be a good application paper in material science with an interesting combination of techniques from machine learning and computer vision. It would be more appropriate to submit this paper to a domain-specific venue rather than to ICLR. Therefore, I cannot justify its acceptance to ICLR in its current format.

Strengths
-----------------------
1.	New domain: The paper addresses an interesting problem in a new domain. Previous work on generative models for molecules have concentrated on molecules with 1D structure (which can be represented as strings) or 2D structure (which can be represented as planar graphs). Generating 3D molecule structures is a challenging and interesting problem.
2.	Efficient data representation: The authors bypass the difficulties associated with modeling sets of 3D points with arbitrary structure by proposing a canonical, voxel-based density representation.
3.	Joint VAE + UNet training: The joint training of the encoder-decoder VAE and the 3D segmentation network results in a decoder that can reconstruct atom locations and atom types, being robust to mistakes in the density map reconstruction.
4.	Qualitative results: There are nice visualizations of the reconstructed molecules and of the effect of the latent variable z. Figure 3 in particular does a great job at elucidating the outputs of the network and the reconstruction quality.

Weaknesses
------------------------
1.	Limited methodological novelty: The methods used in the paper, i.e. the data representation (see detailed comments), the encoder network, the decoder network and the segmentation network are all existing methods without any (or with only minor) modifications.  Their combination also seems straightforward, except for the joint training of the VAE and the segmentation network, which has not been tried before to the best of my knowledge. Networks have not been modified to exploit the intricacies of the new domain/task, such as symmetries, repetition of unit cells, large number of atoms.
2.	Lack of comparison with other methods/baselines: There is no quantitative comparison with alternative baselines or methods or even discussion of such alternatives. It is understandable that the paper addresses a relatively new problem, however the method could be compared to CrystalGAN. Also, there is no justification of the advantages of using a voxel-based density map representation along with 3D convolutions vs, for instance, a graph representation along with graph convolutions (Xie & Grossman, [3]) or a point cloud GAN (Achlioptas).
3.	No quantitative evaluation of the generative capabilities of the network: In the case of drug molecule generation, logP and QED scores  [4,5] are used to evaluate their drug-likeness. Without such scores in the crystal generation domain, it is not easy to judge how good a generated crystal is.  The need for some type of quantitative evaluation is especially important, since it is not as easy to qualitatively judge the quality of crystals obtained by sampling random latent vectors (Fig 5B) as in the case of generated images or text. The only such evaluation presented in the paper is related to the distribution of distances between atoms which seems to be close to their actual distribution in nature.
4.	Presentation of quantitative results: The quantitative results are scattered in text throughout the results section without being summarized in a table. Results about the distance of generated atoms w.r.t to their true location, the predicted atom counts, predicted atom types etc. for the cases of single unit cells and repeated unit cells should be added to a table to complete a rigorous experimental evaluation.

Additional Comments
-------------------------
1.	Lack of novelty: The data representation, which is presented as one of the core contributions of the paper, is not new. Radial basis functions have been used to generate 3D density maps (e.g. [2]) and 3D Convolutional VAEs (Brock2016, [1]) have been employed on density maps (e.g. occupancy maps) on computer vision tasks.
2.	Confusing description of the repeated unit cells case: It is not clear, especially for a general audience, how the repeated unit cells data are generated (in which directions are the unit cells repeated, how many times). It is also not clear how the training routine (conditioning) and the output post-processing (connected components + majority voting) is modified.
3.	After Eq. (2), q is the --approximate-- posterior …

Suggested References
--------------------------

1.	VoxNet: A 3D Convolutional Neural Network for Real-Time Object Recognition
2.	VV-NET: Voxel VAE Net with Group Convolutions for Point Cloud Segmentation
3.	Dynamic Edge-Conditioned Filters in Convolutional Neural Networks on Graphs
4.	Graph Convolutional Policy Network forGoal-Directed Molecular Graph Generation
5.	Grammar Variational Autoencoder
6.	3D U-Net: Learning Dense VolumetricSegmentation from Sparse Annotation
7.	PointConv: Deep Convolutional Networks on 3D Point Clouds


**Experience Assessment:**

I do not know much about this area.

**Review Assessment: Checking Correctness Of Derivations And Theory:**

N/A

**Review Assessment: Checking Correctness Of Experiments:**

N/A

**Review Assessment: Thoroughness In Paper Reading:**

N/A

---

> ### Author Response · Authors · 2019-11-08
> **Thank you for your very detailed review.**
>
> Thank you for spending the time to read the paper carefully and provide very useful feedback. This will be very helpful as we try to revise our manuscript. We wanted to address a few points:
>
> First, thank you for acknowledging that the new domain is a strength of the paper. We hope to inspire more work in an important area and think that by combining physical intuition for input representation and using a VAE/U-Net, this is a useful paper tackling a very important problem in a field related to, but in many ways quite different from, that of drug discovery.
>
> >> Limited methodological novelty
> Regarding the limited methodological novelty: A key problem for a new domain is often the choice of representation. Some times, a new representation and method go hand-in-hand. However, we feel in this case the main hinderance is not existing methods, but instead a representation. Work published since this submission uses a similar representation to ours [1].
>
> >> Lack of comparison with other methods/baselines:
> Comparing to other baselines is a very important aspect of science. Unfortunately, to our knowledge there is no method that produces materials of the complexity we consider. CrystalGAN produces a composition rather than a specific 3D description (that would be needed) and only considers a small subset of material types. Graph networks like those used by Xie and Grossman have been very successful for predicting material properties. However, using this approach to generate the 3-D locations and species of many atoms in space (with a variable number of atoms) is not a problem we are aware of being tackled yet. We will make this more clear in an updated version of the paper. See for example [1] which is a paper that came out after the submission of this work that uses many very similar ideas and also does not perform such evaluations.
>
> >> No quantitative evaluation of the generative capabilities of the network:
> Regarding quantitative evaluation: As you point out, logP and QED scores are commonly used for drug-like molecules. However, these are not applicable for the molecules we are looking at. Additionally, because of the physical constraints, creating materials that relax is quite difficult. Therefore, we report distances as in [2]. We will try to include more metrics in an updated version shortly, but unfortunately there is not yet a community accepted set of criteria as generating valid molecules is already a very difficult problem.
>
> >> Presentation of quantitative results:
> Regarding the presentation of quantitative results— thank you for your feedback. We will restructure the results to be more clear. Thank you!
>
> >>Lack of novelty:
> Regarding the novelty- the use of a 3-D representation  based on the physically meaningful atomic number is something that we found to work well as a possible choice of representation for these 3-D crystal  structures. We feel that the application is important, general, and something that benefits from a discussion as to how to best represent molecules, due to the unique requirements of the field.
>
>
> Thanks again very much for the very constructive comments. We really appreciate the time you spent to provide feedback on this manuscript. We are clarifying the repeated unit cell text and will update the text with the word approximate. Also, thanks for the additional references! :)
>
>
> [1] Inverse design of Solid-State materials via a Continuous Representation. Juhwan Noh, et. al. Matter.
> [2] Symmetry-adapted Generation of 3D point Sets for the Targeted Discovery of Molecules, Gebaueer, Gastegger, and Schütt.

---

> ### Author Response · Authors · 2019-11-14
> **Reviewer 3**
>
> Hello,
> We want to ask if there is anything more we can clarify in the next day or any more comments you may have about the paper?
> Thanks!

---

### Official Review · AnonReviewer1 · 2019-10-28
**Official Blind Review #1**

**Rating:** 3

**Review:**

The authors describe a method to encode and decode the position of atoms in 3-D molecules. An encoder-decoder architecture is used to create a representation of a molecule and to reconstruct the molecule from its representation. Then a second Neural Network segments the output and assigns an atomic number. Prior work on this task has used 1D (SMILES) and 2D (Graph) representations. The authors argues that exploiting 3D structure can create better representations.

As the paper's related work section shows, this is not the first attempt to use 3D structure to create molecular representations. Unfortunately, the paper does not compare their work to prior work on 3D structure representations (e.g Gebaur et al 2019). Also, it is not clear whether the 3D representation is better than 1D or 2D representations especially since there have been many new 1D models that perform very well for tasks like molecular property prediction (For example All SMILES VAE https://arxiv.org/abs/1905.13343 ). I think the community will benefit if the authors perform a comparison with state of the art 1D and 2D models. I think this is a main drawback of this work.

Another concern is that the authors claim that the errors in atomic numbers differ only by 1 or 2. But doesn't this show that the network has not learnt a good representation? Because atoms that differ in atomic number by 1 or 2 will have different valencies and hence exhibit different properties? On the other hand, if the authors can show that the errors in atomic numbers suggest they correspond to similar atoms (may be along the same column in the periodic table), then one can have better confidence that the network has learned a meaningful representation.

**Experience Assessment:**

I have published one or two papers in this area.

**Review Assessment: Checking Correctness Of Derivations And Theory:**

I carefully checked the derivations and theory.

**Review Assessment: Checking Correctness Of Experiments:**

I carefully checked the experiments.

**Review Assessment: Thoroughness In Paper Reading:**

I read the paper at least twice and used my best judgement in assessing the paper.

---

> ### Author Response · Authors · 2019-11-08
> **Regarding the representation of the data**
>
> First,  thank you for your comments on our draft. We are in the process of updating the draft, but want to address a few points.
>
> Regarding Gebauer’s work on SchNet,  their current method works with a smaller subset of the chemical space than is  in the dataset that we use. The specifically state that future work  will explore this direction,  but the increased variation in composition makes this very challenging. Therefore, we focus more on using physical  intuition to try to develop and test a possible data representation.
>
> Regarding a comparison with 1D and 2D methods: this is a very interesting point, and something we thought about quite a bit. For most molecules where ML tools has been successful, SMILES strings or graphs have proved an effective data representation. These tend to be relatively small organic molecules from the QM9 database. There are only a handful of possible species types. In all these cases, a string or graph is generated and there is a mechanism by which this string or graph is “translated” back to a molecule. For crystal structures, the most effective representation  has not been explored. Surely, when a 1D or 2D representation works, it  is far better. For this reason, we found it  very difficult to compare  since these 3D molecules do not have an appropriate SMILES or graph representation that we can use to recover both the 3D geometric arrangement and the composition.  In part, we are hoping to encourage the community to try to begin exploring a more complicated chemical space (similar to G-SchNet). Since we submitted, two new papers have come out in this area [1,2].  Notably, [2] uses ideas very similar to the ones we presented but also does not provide a clear baseline, due to the novelty of the domain. We will add a section in the supplement and main text that more clearly addresses this.
>
> >> Another concern is that the authors claim that the errors in atomic numbers differ only by 1 or 2.
> This is a very real concern, and something that we thought about considerably. Originally, we tried encoding and decoding group/period rather than atomic number, and found (understandably) improved accuracy. In fact, we found we were able to reconstruct group with just over 93% accuracy. However, for many applications (such as random structure searching) we felt that predicting atomic number and then introducing potential post-processing steps was a preferable alternative. We will add our results on group to the main text.
>
> Thanks  again for reading the paper and providing very helpful feedback. We will post an updated manuscript soon, but hope that we have addressed some of your concerns.
>
> [1] “Generating valid Euclidean distance matrices” Moritz Hoffmann, Frank Noé arXiv:190.03131
> [2] Inverse design of Solid-State materials via a Continuous Representation. Juhwan Noh, et. al. Matter.

---

> ### Author Response · Authors · 2019-11-14
> **Review 1**
>
> Hello,
> We want to ask if there is anything more we can clarify in the next day or any more comments you may have about the paper?
> Thanks!

---

### Author Response · Authors · 2019-11-10
**Revision # 1**

We uploaded a new version of the paper that we think is a large improvement. We would like to thank the reviewers for very helpful feedback. Below, we outline the specific changes made:
• The major concerns of Reviewers 1 & 3 was a comparison with existing methods and the choice of representation. We added a section in the main text to specifically address this, and also clarified various other parts of the main text. We tried to make it clear that existing 1-D and 2-D methods are not appropriate for the task we consider.
• Based on the comments of Reviewer 2, we have added M and S to Fig. 2. Thank you for this suggestion. We also added a new section in the supplement on the choice of grid size and spacing.
• Based on the suggestion of Reviewer 3, we have tried to make the results section more streamlined. We have moved a few pieces around and added more descriptive text when needed. We also added more section numbers, to help clarify the flow of the text.
• We expanded Table 1 to include species information, at the suggestion of Reviewer 3.
• We made minor changes, including clarifying that q is the approximate posterior.
• Added a few new references (Hoffmann  & Noé), those suggested by Reviewer 1 (Alperstein, Cherkasov, and Rolfe) and those suggested by Reviewer 3.
• We improved the clarity throughout the text of the paper.
[EDIT]
• To try to further improve the understanding of the current state of the randomly generated samples (due to the difficult in comparing to current baselines), we added a new Figure (Fig. S7) that shows more random samples of varying complexity. We will release pre-trained models and code after all deadlines.

---

### Decision · Program_Chairs · 2019-12-19

**Decision:**

Reject

**Comment:**

This paper presents an encoder-decoder based approach to construct a compressed latent space representation of each molecule. Then a second neural network segments the output and assigns an atomic number. Unlike previous works using 1D or 2D representations, the proposed method focuses on the 3D representations.

The reviewers have several major concerns. Firstly, the novelty of the paper seems to be limited as the proposed method mainly use the existing techniques. Secondly, there is no clear baseline to compare with. Finally, there is no clear quantitative results to measure the proposed method. The rebuttal did not well address these problems.

Overall, this paper did not meet the standard of ICLR and I choose to reject the paper.